# Treatment and Prevention of Cancer-Associated Thrombosis in Frail Patients: Tailored Management

**DOI:** 10.3390/cancers11010048

**Published:** 2019-01-07

**Authors:** Florian Scotté, Pauline Leroy, Mathilde Chastenet, Laure Aumont, Vidal Benatar, Ismaïl Elalamy

**Affiliations:** 1Department of Medical Oncology and Supportive Care. Hôpital Foch, 92150 Suresnes, France; p.leroy@hopital-foch.com (P.L.); chastenet.mathilde@gmail.com (M.C.); l.aumont@hopital-foch.com (L.A.); 2Heathics Clinical Consultants. 111 rue des Tennerolles, 92210 Saint-Cloud, France; vidal.benatar@heathics.com; 3Department of Hematology, Hôpital Tenon, Hôpitaux Universitaires de l’Est Parisien, Sorbonne Université, INSERM UMR S938, 75012 Paris, France; ismail.elalamy@aphp.fr

**Keywords:** frailty, cancer, cancer-associated thrombosis, anticoagulants

## Abstract

Advanced age is one of the major determinants of frailty in patients with cancer-associated thrombosis. However, multiple other factors contribute to frailty in these patients. The identification of frailty in patients with cancer-associated thrombosis is critical as it influences the complexity of the anticoagulant treatment in this population at high risk of venous thromboembolism and bleeding. Factors that contribute to frailty in patients with cancer-associated thrombosis include age, type of cancer, comorbidities such as chronic kidney disease, poly-pharmacotherapy, treatment compliance, cognitive impairment, anemia, thrombocytopenia, mobility, nutritional status, Eastern Cooperative Oncology Group grade, risk of falls, and reduced life expectancy. In the absence of specific clinical studies current anticoagulant treatment guidelines for the management are not fully applicable to frail patients with cancer. The anticoagulant treatment should therefore benefit from a tailored approach based on an algorithm that takes into account the specificities of the malignant disease.

## 1. Introduction

According to the American Medical Association (AMA), the term “frailty” characterizes “the group of patients that presents the most complex and challenging problems to the physician and all health care professionals”, because they have a higher susceptibility to adverse outcomes such as mortality and institutionalization [1]. In the United States, frailty is estimated to occur in 15–25% of community-dwelling older adults (≥65 years and older) [2]. For more than 30 years no clear consensus among clinicians has been reached on how to define frailty. Frailty is usually designated as a multidimensional syndrome that includes loss of energy, physical ability, cognition, and health. Several tools have been developed to provide predictive information on the risk of death or the need to implement specific measures for the management of frail patients. Among the observed heterogeneity of viewpoints, a definition focusing on five domains with the corresponding criteria including nutritional status (loss of bodyweight), energy (exhaustion), physical activity (leisure time activity), and strength (grip strength) is frequently used [3]. This allows defining the frail phenotype and identifying older persons at elevated risk for numerous adverse outcomes.

Regarding the use of anticoagulants several situations at risk representing multiple challenges for the management of frail patients have been considered. They include chronic kidney disease (CKD), underweight or malnutrition, falls, cognitive impairment, multi-medication, pregnancy and cancer [4]. All these situations may apply to patients with cancer to define frailty.

The identification of frailty in patients with cancer-associated thrombosis (CAT) is critical since malignancy is associated with multiple factors, making these patients highly vulnerable, especially in the context of venous thromboembolism. Anticoagulant treatment is quite complex since patients with cancer are at higher risk of a first venous thromboembolism (VTE), VTE recurrence, and bleeding compared to non-cancer patients [5]. This includes either patients with CAT requiring anticoagulant treatment or patients at high risk of thromboembolism eligible for primary thromboprophylaxis. The evaluation of the benefit-risk ratio of anticoagulant treatment is therefore of particular importance in frail patients such as the very elderly.

There are no specific anticoagulant treatment guidelines for the management of elderly frail patients with cancer and their management rests on an interpretation of existing guidelines. The aim of this review was to identify main contributors to frailty in patients with cancer at risk of VTE and to discuss a decision-making algorithm for the management of the anticoagulant treatment in these patients.

## 2. Risk of Thrombosis and Bleeding in Patients with Cancer

Patients with cancer are at higher risk of VTE compared to patients without cancer. The Khorana score is validated to predict the risk VTE in outpatients with cancer [6] (Table 1). This score was shown to be predictive of in-hospital, symptomatic VTE development in cancer patients who are hospitalized for medical reasons [7]. The Modified Ottawa Score (MOS) (Table 2) was set to predict the risk of VTE recurrence in patients with CAT [8]. Even though it was validated in outpatients with cancer-associated VTE, the MOS failed to predict VTE recurrence in hospitalized patients with CAT [9,10] thus suggesting that a new scoring system is required in this setting. Nevertheless none of existing scoring systems take into account the specificity of frail patients with cancer, especially the elderly.

There is no validated scoring system to predict the risk of bleeding in frail patients with cancer receiving anticoagulant treatment. The American College of Chest Physicians (ACCP) has established a list of evidence-based risk factors for bleeding and estimated risk of major bleeding [11]. Taking into account the ACCP list, frail patients with CAT accumulate a number of some relevant risk factors for bleeding such as age ≥75, cancer, metastatic cancer, chronic kidney disease (CKD), poor anticoagulant control, co-morbidity and reduced functional capacity, and frequent falls, knowing that high risk of bleeding (total risk of 12.8% during up to 3 months of anticoagulation) is defined by ≥2 risk factors [11].

Very elderly patients were shown to be eligible for anticoagulant treatment with vitamin K antagonists (VKA) without an increased risk of bleeding [12]. The increased risk of bleeding in patients with cancer is well established. The anticoagulant-associated bleeding risk including major bleeding is higher in cancer patients compared to non-cancer patients [13].

Several studies have documented the risk of bleeding in CAT patients treated with anticoagulants. The rate of major bleeding in cancer patients receiving anticoagulant treatment has been reported as between 4.1% and 12.4% [14,15]. Among the 3805 patients with CAT enrolled in the Computerized Registry of Patients with Venous Thromboembolism (Registro informatizado de la enfermedad tromboembólica en España) called RIETE cohort registry [16], patients with immobility (OR: 1.8; 95% CI: 1.2–2.7), metastases (OR: 1.6; 95% CI: 1.1–2.3), recent bleeding (OR: 2.4; 95% CI: 1.1–5.1), or with creatinine clearance <30 mL/min (OR: 2.2; 95% CI: 1.5–3.4), had an increased incidence of major bleeding. In the Comparison of Low-molecular-weight heparin versus Oral anticoagulant therapy for the prevention of recurrent venous Thromboembolism in patients with cancer (CLOT) study ([17] the rate of major bleeding was 6% (19 of 338) and 4% (12 of 335) with the low-molecular-weight heparin (LMWH) dalteparin and warfarin, respectively while in the Comparison of Acute Treatments in Cancer Hemostasis (CATCH) study [18] the rate of major bleeding with LMWH (tinzaparin) and warfarin was 2.7% (12 of 449) and 2.4% (11 of 451), respectively. In a meta-analysis of five studies (N = 19,060) the pooled incidence rates of clinically relevant bleeding (CRB), which includes both major and clinically relevant non-major bleeding, in the sub-group of CTA patients treated with direct oral anticoagulants (DOAC) was 15% (95% CI 12–18]) [19]. Two recent randomized-control studies have shown that in CAT patients treated with rivaroxaban or edoxaban the risk of major bleeding ranged between 6% and 7% [20,21]. In the Hokusai VTE-cancer trial [20] the rate of CRB was 18.6% (97 of 522) and 13.9% (73 of 524) (hazard ratio, HR 1.40 (95% CI 1.03–1.89)) in CAT patients treated with edoxaban and dalteparin, respectively. 

Factors associated with the risk of bleeding include thrombocytopenia caused by chemotherapy [22] while age >75 was identified as significant risk factor of CRB (RR 1.79 (95% CI 1.18–2.70) *p* = 0.013) [23].

Risk of bleeding [24], bleeding risk stratification models appear to have little accuracy in very elderly patients with VTE [25].

## 3. Factors Contributing to Frailty in Patients with Cancer-Associated Thrombosis

Frailty in patients with cancer results from the combination of multiple factors that may increase the risk of bleeding or thromboembolism. Frailty in patients with cancer results from overlapping domains of aging, Eastern Cooperative Oncology Group (ECOG) status, type of cancer, poly-pharmacotherapy, cognitive impairment, blood disorders, and reduced life expectancy (Table 3).

### 3.1. Aging

Cancer and frailty are associated with advanced age. Frailty in community-dwelling adults increases with age, affecting 11% of the elderly over the age of 65 years and 25% of those over the age of 85 years [26].

Aging is a supplementary factor that contributes to frailty in patients with CAT, making the management of anticoagulant treatment complex. The use of concomitant anti-cancer therapies (chemotherapy, hormones, immuno-modulatory or anti-angiogenic drugs), central venous catheter (CVC) placement, and invasive cancer surgery further increase the thrombotic risk and expose patients to potential drug interactions. The risk of VTE recurrence is usually higher in patients with advanced-stage cancer receiving chemotherapies and sub-cutaneous growth factors [27].

Elderly patients (aged > 75) with cancer are at particularly high risk of bleeding not due only to age and renal dysfunction, but also to the more frequent side effects from cancer therapy and a generally frailer situation [28].

### 3.2. Eastern Cooperative Oncology Group 

The ECOG scale of performance status is a consistent and convenient manner for measuring the impact of cancer on the patient’s capabilities (Table 4) [29]. A high ECOG grade of 3–4 may result from advanced age, cancer progression, malnutrition, or falls that compromise patient’s autonomy and contribute to frailty.

### 3.3. Cancer Disease

Patients with recently diagnosed active cancer are at much higher risk of VTE recurrence and bleeding compared to patients with only a history of cancer [30]. In this latter category, the incidence of VTE recurrence and bleeding is similar to that reported in patients without cancer [31].

VTE is one of the most frequent complications in patients with metastatic pancreatic ductal adenocarcinoma (PDAC), with an incidence of 20–35% [32]. The etiology is multifactorial and involves general as well as tumor specific factors. The routine use of prophylaxis for VTE is not recommended in ambulatory patients due to the variable bleeding risk, and unclear results from clinical trials. However, it is recommended to periodically evaluate the risk, discuss symptoms and potential risks with the patient, and consider using prophylaxis with low molecular weight heparin (LMWH) in patients with high risk of TED (Khorana index ≥3) without risk of bleeding. These patients require close monitoring [33].

Primary pharmacological prophylaxis of VTE with LMWH is indicated in patients with locally advanced or metastatic pancreatic cancer treated with systemic anticancer therapy and who have a low bleeding risk (grade 1B) [34,35].

Lung cancer is the most common malignancy worldwide and the leading cause of cancer-related death [36]. It is associated with a high risk of VTE which may be further increased by the use of antineoplastic chemotherapy [37]. Several scoring systems to predict VTE in patients with lung cancer have been proposed, including the Khorana score (Table 1) [6,38], thePROphylaxis of ThromboEmbolism during CHemoTherapy (PROTECHT) score, and the CharitéONKOlogie (CONKO) score. However, these scores have been undermined or not validated [39,40,41]. The Comparison of Methods for thromboembolic risk assessment with clinical Perceptions and AwareneSS (COMPASS)-cancer-associated thrombosis (COMPASS-CAT) scoring system seems to be most predictive of the risk of VTE development as it includes patient and cancer-related risk factors and comorbidities as well as the oncological treatment administered [42].

Hematologic malignancies are mostly diagnosed in patients of older age [43], while adults with blood cancer aged >75 years represent only 11% of the enrollment in clinical trials [44]. Supporting data are therefore limited to assess frailty in patients with hematologic malignancies.

In patients with multiple myeloma (MM) the International Myeloma Working Group (IMWG) used a simplified geriatric assessment (GA) tool based on age, co-morbidities, and activities of daily living (ADL). A score was developed that classifies patients as fit (score = 0, 39%), intermediately fit (score = 1, 31%), and frail (score ≥ 2, 30%) [45]. The IMWG frailty score was predictive of mortality, treatment discontinuation, and non-hematologic toxicities. MM has one of the highest risks of thrombosis among all cancers due to disease-related pathological changes and treatment [46,47]. Thalidomide and lenalidomide (IMIDs) are well known to be associated with increased risk of thrombosis [48].

In patients with non-Hodgkin’s lymphoma (NHL), GAs utilizing age, ADL, co-morbidities, and geriatric syndromes were shown adequate to assess frailty and to identify patients fit for curative intent chemotherapy [49]. Elderly patients with acute myelogenous leukemia are among the most vulnerable patient population, with the most diagnoses and deaths in those aged 65 years and older [50]. 

### 3.4. Comorbidities

Chronic kidney disease (CKD) is more frequent in cancer patients due to several factors such as dehydration or older age-related decrease in the glomerular filtration rate. Renal function may be further decreased by anti-cancer treatments such as cytotoxic drugs, tyrosine kinase inhibitors and androgen deprivation therapy [51,52], and also by non-anti-cancer drugs. CKD is an independent risk factor for bleeding and represents a critical therapeutic challenge for the choice of an anticoagulant as up to 50% of patients with cancer may have renal dysfunction [53]. The risk of major bleeding is significantly increased in patients with creatinine clearance below 30 mL/min and metastatic cancer [54] as well as the risk of fatal bleeding [55] since anticoagulants may accumulate with the compromised renal elimination. In patients with cancer, the risk of major bleeding is correlated to the degree of renal impairment [56]. In the CATCH trial, CKD was associated with a statistically significant increase in recurrent VTE (RR 1.74 (95% CI: 1.06; 2.85)) and major bleeding (RR 2.98 (95% CI: 1.29; 6.90)) in CAT patients with CKD compared to patients without CKD, demonstrating the fragility of these patients [57].

Hepatic function may be severely reduced as a result from organ metastasis or hepatotoxic side effects of anticancer treatment

Cancer patients frequently suffer from infections leading to antibiotic or anti-mycotic treatments causing additional gastrointestinal side-effects such as vomiting, gastric atrophy or diarrhea. All these features must be taken into account since they have a direct implication on pharmacokinetics/pharmacodynamics relationship (PK/PD) properties of oral anticoagulants [30].

### 3.5. Poly-Pharmacotherapy

Metabolic pathways involving cytochrome P450 3A4 (CYP 3A4) and P-glycoprotein (Pgp) must be taken into account in cancer patients as several cancer treatments and anticoagulants are metabolized through common pathways. Drug–drug interactions may modify anticoagulants pharmacodynamics and pharmacokinetics, resulting in an increase of either thrombotic or bleeding risks [58].

The use of multiple concomitant treatments significantly increases the bleeding risk as shown in a trial in 18,201 patients with atrial fibrillation treated with anticoagulants [59].

### 3.6. Cognitive Impairment

Compliance to the treatment is a challenge with the use of oral anticoagulants in patients with cancer. The key consequence of cognitive impairment is compromising the oral anticoagulant treatment compliance in outpatients who are not regularly monitored. Patients may forget to take their anticoagulants, leading to an increased thrombotic risk, or take the treatment repeatedly, as they forget the initial administration, leading to overdosing with increased bleeding risk. Cognitive impairment was shown to be associated with an increased risk of VTE and bleeding in patients with atrial fibrillation [60]. Chen et al. documented a rate of 34.1% of noncompliance with warfarin therapy among 7612 patients at high risk of VTE (including cancer patients) [61]. Noncompliant patients had a three times greater risk of VTE recurrence than compliant patients (hazard ratio, HR = 3.01).

### 3.7. Blood Disorders

The bleeding risk is also increased because of chemotherapy-induced thrombocytopenia and hepatic or brain metastases. A previous episode of major bleeding in the last two months or the presence of intracranial or visceral tumor increases the risk of major bleeding [15] as does a platelet count fall below 50,000 per μL secondary to chemotherapy [62].

Conversely, thrombocytosis with platelet count >400,000/mL [63] or ≥350,000/mL [6] is associated with an increased risk of VTE in patients with cancer.

Anemia poses a difficult challenge since cancer patients with anemia receiving anticoagulants were shown to have a higher rate of major bleeding compared to non-cancer patients [64] while aplastic anemia was shown to increase the risk of VTE [65].

## 4. Management of Anticoagulants in Frail Patients with Cancer-Associated Thrombosis

Anticoagulation is particularly complex in frail elderly patients who present with additional risk factors that affect both the efficacy and the safety of VTE or treatment. There is no formal guideline for the management and treatment of frail patients with CAT, as major clinical trials generally do not include elderly frail patients. The impact of frailty was analyzed in a literature review of 1204 hospitalized elderly patients (mean age of 85 ± 6) with atrial fibrillation (AF) showing that the use of anticoagulants was low in these patients, thus suggesting that frail elderly patients are less likely to receive oral anticoagulants [66].

A decision-making algorithm has been developed for the use of anticoagulants in patients with AF [67]. Given the similarities regarding age and frailty in AF and cancer patients we found appropriate to adapt the AF algorithm to patients with cancer taking into account specific frailty contributing factors in relation with malignancy (Figure 1).

Data on the use of anticoagulants in frail patients with cancer are limited. Elderly represent a small proportion of patients included in large randomized-control trials (RCTs) comparing vitamin K antagonists (VKAs) to LMWH or comparing LMWH to direct oral anticoagulants (DOACs) for the long-term treatment or prophylaxis of CAT.

In RCTs, LMWH was shown to be superior to VKA in reducing the incidence of VTE recurrence [17,68,69,70,71] and in the primary prevention of VTE in patients with cancer [72]. DOACs were shown to be non-inferior to LMWH regarding efficacy while they were associated with a significant excess of bleeding compared to LMWHs [20,21].

As shown in Table 3 some critical aspects must be taken into account for the choice of an anticoagulant in frail patients with cancer. The choice of the anticoagulant is driven by the balance between the efficacy against thromboembolism and the safety due to the risk of bleeding in a highly vulnerable population.

Cognitive impairment in frail elderly patients may compromise treatment adherence with oral anticoagulants. Unless close patient monitoring is possible, injectable LMWH may be preferred to ensure treatment compliance.

Frail patients with cancer treated with either VKA or DOAC are highly likely to experience pharmacokinetic drug–drug interactions due to the need for concomitant antineoplastic drugs, drugs to treat comorbidities, drugs to relieve symptoms from cancer, and drugs to treat cancer treatment-induced adverse events which are frequently either P-gp and CYP 3A4 inhibitors or inducers. Also, drug–drug interactions may be associated with decreased anticoagulation effects during the concomitant use of DOACs with anti-neoplastic drugs which are CYP 3A4 or P-gp inductors, while the bleeding risk may be increased with the concomitant use of DOACs and CYP 3A4 or P-gp inhibitors [58]. Multiple pharmacotherapies (more than five drugs) may therefore create confusion and increase the risk of thromboembolic or the risk of bleeding in insufficiently monitored outpatients. When possible but taking into account patient’s convenience LMWH should be preferred.

The anticoagulant treatment should be discussed and maintained as long it does not compromise the antineoplastic treatment.

Even though trials suggest that DOACs are of comparable efficacy with respect to LMWH in preventing VTE recurrence, they are associated with an excess of bleeding in patients with cancer [20,21,73]. DOACs should therefore be avoided in frail patients at high risk of bleeding, making LMWH the first choice in these patients, especially in the presence of gastrointestinal cancer [74].

In case of renal impairment the accumulation of the anticoagulant effect should be avoided. Both LMWH and DOACs may be used in patients with CrCl >15 mL/min, with the exception of the DOAC dabigatran which should not be used in patients with CrCl <30 mL/min.

## 5. Conclusions

The management of frail patients with cancer is extremely complex due to the lack of consensus in the definition of frailty, the paucity of evidence data based on clinical trials, and the absence of specific clinical guidelines.

We have attempted to clarify the clinical context of these patients and to simplify the therapeutic decision-making with the development of the proposed decision tree (Figure 1).

Besides advanced age, the most significant factors to be taken into account when assessing frailty patients with cancer are ECOG reflecting patient’s general condition (nutritional status, body weight and mobility), the type of cancer especially gastrointestinal (pancreas), lung, and multiple myeloma, and comorbidities. Poly-pharmacotherapy and cognitive impairment leading to drug–drug interactions and/or compromised treatment adherence, blood disorders, risk of falls, and reduced life expectancy must also be considered on a case by case approach. Frailty in patients with cancer results from the accumulation and overlapping of multiple factors which make this population highly vulnerable and the decision-making particularly complex for the clinician (Figure 2). The management of frail CAT patients suffers from further complexity due to insufficient evidenced-based data to guide the therapeutic strategy since these patients, especially the oldest, are under-enrolled and understudied in clinical trials [44].

The identification and assessment of frailty in patients with CAT is therefore essential to optimize the anticoagulant treatment while maintaining quality of life. Managing anticoagulants in frail patients with CAT is highly complex and represents a major challenge as these patients are at high risk of bleeding due to the accumulation of bleeding risk factors. The anticoagulant treatment should minimize the bleeding risk while maintaining the antithrombotic efficacy and preserving the efficiency of antineoplastic drugs which may be altered by drug–drug interactions in the context of multiple therapies and co-morbidities. Given the multiple challenges related to the condition of frail patients with CAT, parenteral anticoagulation seems preferable to oral anticoagulation in patients receiving cytotoxic drugs with severe gastrointestinal and/or hematological side effects or suffering gastrointestinal malignancy. Treatment or prophylaxis of thromboembolic disease in frail patients with cancer therefore requires a carefully tailored approach. The complexity of frailty assessment in patients with CAT requires establishing scoring methods adapted to each specific situation to guide patient management.

The development of a scoring system likely to predict both the VTE and bleeding risks to adjust anticoagulant treatment is warranted.

## Figures and Tables

**Figure 1 cancers-11-00048-f001:**
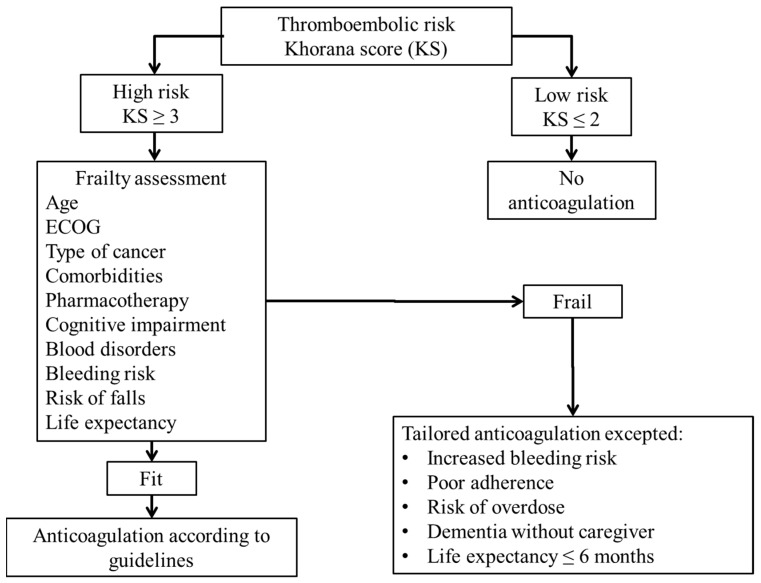
Decision algorithm for anticoagulant treatment in frail patients with cancer. ECOG: Eastern Cooperative Oncology Group.

**Figure 2 cancers-11-00048-f002:**
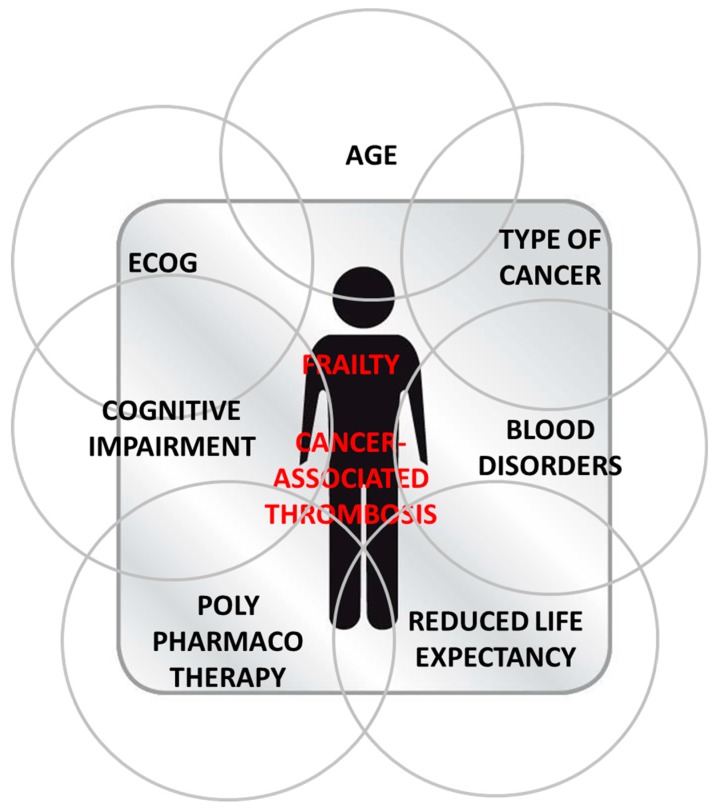
Complex interaction and overlapping factors contributing to frailty in patients with cancer-associated thrombosis.

**Table 1 cancers-11-00048-t001:** Predictive model for venous thromboembolism [6]

Patient Characteristics	Risk Score
Site of cancer	
Very high risk (stomach, pancreas)	2
High risk (lung, lymphoma, gynecologic, bladder, testicular)	1
Pre-chemotherapy platelet count ≥350 × 109/L	1
Hemoglobin level <10 g/dL or use of red cell growth factor	1
Pre-chemotherapy leukocyte count >11,000/mm^3^	1
Body mass index ≥35 kg/m^2^	1
Clinical probability for VTE*	
High-risk score	≥3
Intermediate-risk score	1–2
Low-risk score	0

* VTE = venous thromboembolism.

**Table 2 cancers-11-00048-t002:** Ottawa Score for recurrent VTE risk in cancer-associated thrombosis.

Patient Characteristic	Risk Score
Female	1
Lung cancer	1
Breast cancer	−1
TNM * stage I	−2
Previous VTE	1
Clinical probability for VTE recurrence	
Low (≤0)	−3 to 0
High (≥1)	1 to 3

VTE = venous thromboembolism; * TNM = tumor–nodes–metastasis staging system (for solid tumors only).

**Table 3 cancers-11-00048-t003:** Factors contributing to frailty in patients with cancer-associated thrombosis (CAT).

Factors	Assessment	Impact on Patient Management
Age	Patients aged ≥ 75	Frailty assessment
ECOG Nutritional status Mobility	Loss of body weight Swallowing disorders Monitoring barriers	No food interaction with LMWH compared to oral anticoagulants LMWH preferred in case of severe swallowing disorders Oral anticoagulants more practical than LMWH
Type of cancer	Pancreas Multiple myeloma	LMWH for VTE prophylaxis and treatment LMWH if concomitant use of IMiDs
Comorbidities	Renal impairment Hepatic impairment	LMWH or DOAC in patients with CrCl <15 mL/min (<30 mL/min for dabigatran) LMWH preferred to oral anticoagulants
Poly-pharmacotherapy Antineoplastic treatment Supportive therapies	Number of drugs Increased thromboembolic events with IMiD in patients with myeloma Drug-drug interactions	Prioritize antineoplastic treatment in patients receiving ≥5 drugs. LMWH on a case-by-case basis LMWH preferred to oral anticoagulants
Cognitive impairment	Poor treatment compliance	No oral anticoagulants unless systematic follow-up visits LMWH to be preferred for adherence purposes
Blood disorders Anemia Thrombocytopenia	Increased risk of VTE Increased bleeding risk	
Risk of falls		LMWH or oral anticoagulants
Reduced life expectancy	To be considered	Consider avoiding anticoagulants in case of life expectancy ≤6 months

ECOG = Eastern Cooperative Oncology Group; LMWM = low-molecular-weight heparin; DOAC = direct oral anticoagulant; CrCl = creatinine clearance; VTE = venous thromboembolism; IMiD = immunomodulatory drugs; VTE = venous thromboembolism.

**Table 4 cancers-11-00048-t004:** ECOG performance status (adapted from Oken et al.) [29].

Grade	ECOG Performance Status
0	Fully active, able to carry on all pre-disease performance without restriction
1	Restricted in physically strenuous activity but ambulatory and able to carry out work of a light or sedentary nature, e.g., light house work, office work
2	Ambulatory and capable of all self-care but unable to carry out any work activities; up and about more than 50% of waking hours
3	Capable of only limited self-care; confined to bed or chair more than 50% of waking hours
4	Completely disabled; cannot carry on any selfcare; totally confined to bed or chair

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
