# Peer review of "Treatment and Prevention of Cancer-Associated Thrombosis in Frail Patients: Tailored Management"

_cancers, 2019, doi:10.3390/cancers11010048_

Round 1

Reviewer 1 Report

 The authors tried to identify main contributors to frailty in patients with cancer at risk of VTE and to discuss a decision-making algorithm for the management of the anticoagulant treatment.

The paper provides very interesting review. But from the reviewer point of view, there are many issues that should be made clear.

It is hard to recognize the figure 1 which is most important in this article.

References is not numbered in order of appearance in the text.

So it is very hard to understand the contents of this article.

Authors should read the Cancer’s instructions for authors carefully.

[Abbreviations should be defined in parentheses the first time they appear in the abstract, main text, and in figure or table captions and used consistently thereafter.]

[References: References must be numbered in order of appearance in the text (including table captions and figure legends) and listed individually at the end of the manuscript.]

Line 85-90, each clinical studies (RIETE, CLOT, CATCH) need references.

Abbreviations

The authors should arrange how to use abbreviation.

Abstract: Line 15, CAT should be defined in parentheses.

Line 89/90, LMWH appears in a sentence first without definitions.

Line 144, LMWH appears in a sentence. Line 243, LMWH appeared twice.

Line 243, Abbreviation LMWH appears in a sentence twice

Line 92/97, CRB appears in a sentence twice.

In Fig.1, some characters are unclear.

The references during the text should be expressed with a number, and the The authors must enumerate quotation sequentially.

Author Response

Thank you for your useful comments.

References and   abbreviations have been corrected.

The figure 1 was simplified and does not require further description in the section:"Management of anticoagulants in frail patients with cancer-associated thrombosis"

Reviewer 2 Report

The manuscript by Scotté and colleagues is a narrative review of the main contributors to frailty in patients with cancer at risk of venous thromboembolism (VTE).  Overall, the article is potentially interesting.  However, there are few issues that need to be taken into consideration.

The first resides in a relative lack of novelty, as a recent review on the same topic (J Med Vasc. 2018 Sep; 43(5):302-309. PMID: 30217344) was recently published in PubMed by Elalamy et al. To be honest, we must recognize that the latter is focused on various at-risk situations: chronic kidney disease (CKD), underweight or malnourishment, falls, cognitive impairment, polypharmacy, cancer and pregnancy. Here, a whole section (from line 105 to line 226, approximately half the manuscript) is devoted to the description of factors contributing to frailty “in patients with CAT”, where aging and cancer (but authors are not specifically focusing on cancer?) are included along with ECOG status, co-morbidities (CKD), poly-pharmacotherapy, cognitive impairment, blood disorders and reduced life expectancy.

The second issue concerns the Author statement that “The aim of this review was to …. discuss a decision-making algorithm for the management of the anticoagulant treatment in these patients” (lines 55-56).  However, I couldn’t find any reference to the use of such algorithm and Figure 1, reporting an Authors’ proposal, is not even discussed.  Talking about the bleeding risk in fragile patients, this is not adequately addressed and some important references are not discussed (e.g. Circulation. 2011 Aug 16;124(7):824-9. doi: 10.1161/CIRCULATIONAHA.110.007864). Although I agree with the Authors’ conclusion that treatment or prophylaxis of VTE in frail cancer patients requires a careful tailored approach, it is not clear how the authors propose to do that.

Third, the English form should be revised and the manuscript should be carefully checked for typos and grammar inconsistencies.  Also, Authors should carefully avoid any text redundancy (as in the case of poly-pharmacotherapy)

Minor

Abbreviations should be defined at first appearance (line 46, VTE)

Line 134: Sentence should be clarified.

Line 138: Thromboembolic disease (TED) here refers to VTE

Line 151: by the is duplicated in the sentence

Line 153: the correct reference for the Khorana score is Blood 2008

Line 222: Authors should take into consideration to refer also to the cutoff ≥350,000 proposed in the Khorana score, and possibly discussed

Lines 255-256: Are the Authors sure that “injectable LMWH should be preferred to ensure treatment compliance”?

Author Response

Thank you for your comments

Regarding your comment on Elalamy publication, we added the sentence indicating that all the situations described in the publication may apply to cancer patients in order to define frailty, wish is the focus of this manuscript. I.Elalamy concurs with this sentence (he is a co-author of our submitted manuscript)

Thank you for the Poli D publication. We have cited it (Circulation 2011) wish indicates that very elderly patients are eligible to VKA without excess bleedings (corresponding statement is added in the manuscript).

All minor objections have been addressed and the English form revised

Reviewer 3 Report

In this review Scotté et al., summarize the current knowledge about factors that contribute to frailty in patients with cancer-associated thrombosis, suggesting the need for a tailored approach when proceeding with anticoagulant treatments. The review is well written, comprehensive and timely. I envision some minor points to be addressed:

1) I would like to suggest to avoid abbreviations in the abstract and in the section heads.

2) Taking into account that the Journal is of interest to an international readership, in the introduction it would be better to extend analysis of epidemiology of frailty by citing data form countries worldwide, when available.

3) Authors should add a table summarizing data from studies about risk of bleeding in CAT patients (pag.3, lines 82-99).

4) Figure 1 is poorly presented. Please, change.

Author Response

Thank you for your useful comments

1) Abbreviations have been adpated as suggested

2) Even the high heterogeneity and the lack of consensus on definitions for frailty, reliable epidemiology data are lacking especially for patients with cancer

3) We consider that developing a table on bleeding risk would require an extensive review wish may justify a further publication. We thank the reviewer for this very interesting suggestion. 

4) The figure 1 has been changed and simplified to improve clarity

Round 2

Reviewer 1 Report

The manuscript has been revised well. I think this manuscript will be acceptable after one modification has been done.

In figure 1, The word [Age] is indistinct.

Reviewer 2 Report

The manuscript has been revised and all points raised have been satisfactorily addressed.  At present I can find nothing of substance to comment upon.